# Multiplex Assay for Simultaneous Detection of Antibodies against Crimean-Congo Hemorrhagic Fever Virus Nucleocapsid Protein and Glycoproteins in Ruminants

Alexis C. R. Hoste,[a,b] Igor Djadjovski,[c] Miguel Ángel Jiménez-Clavero,[d,e] Paloma Rueda,[a] John N. Barr,[b] Patricia Sastre[a]

aEurofins-Inmunología y Genética Aplicada S.A. (Eurofins-INGENASA S.A.), Madrid, Spain
bSchool of Molecular and Cellular Biology, University of Leeds, Leeds, United Kingdom
cSs. Cyril and Methodius University in Skopje, Faculty of Veterinary Medicine, Skopje, North Macedonia
dCentro de Investigación en Sanidad Animal (CISA), Instituto Nacional de Investigación y Tecnología Agraria y Alimentaria (INIA-CSIC), Valdeolmos, Spain
eCIBER of Epidemiology and Public Health (CIBERESP), Madrid, Spain

**ABSTRACT** Crimean-Congo hemorrhagic fever virus (CCHFV) is a widespread tick-borne zoonotic virus that causes Crimean-Congo hemorrhagic fever (CCHF). CCHF is asymptomatic in infected animals but can develop into severe illness in humans, with high case-fatality rates. Due to complex environmental and socio-economic factors, the distribution of CCHFV vectors is changing, leading to disease occurrence in previously unaffected countries. Neither an effective treatment nor a vaccine has been developed against CCHFV; thus, surveillance programs are essential to limit and control the spread of the virus. Furthermore, the WHO highlighted the need of assays that can cover a range of CCHFV antigenic targets, DIVA (differentiating infected from vaccinated animals) assays, or assays for future vaccine evaluation. Here, we developed a multiplex assay, based on a suspension microarray, able to detect specific antibodies in ruminants to three recombinantly produced CCHFV proteins: the nucleocapsid (N) protein and two glycoproteins, $G_N$ ectodomain ($G_N$e), and GP38. This triplex assay was used to assess the antibody response in naturally infected animals. Out of the 29 positive field sera to the N protein, 40% showed antibodies against $G_N$e or GP38, with 11 out of these 12 samples being positive to both glycoproteins. To determine the diagnostic specificity of the test, a total of 147 sera from Spanish farms free of CCHFV were included in the study. This multiplex assay could be useful to detect antibodies to different proteins of CCHFV as vaccine target candidates and to study the immune response to CCHFV in infected animals and for surveillance programs to prevent the further spread of the virus.

**IMPORTANCE** Crimean-Congo hemorrhagic fever virus (CCHFV) causes Crimean-Congo hemorrhagic fever, which is one of the most important tick-borne viral diseases of humans and has recently been found in previously unaffected countries such as Spain. The disease is asymptomatic in infected animals but can develop into severe illness in humans. As neither an effective treatment nor a vaccine has been developed against CCHFV, surveillance programs are essential to limit and control the spread of the virus. In this study, a multiplex assay detecting antibodies against different CCHFV antigens in a single sample and independent of the ruminant species has been developed. This assay could be very useful in surveillance studies, to control the spread of CCHFV and prevent future outbreaks, and to better understand the immune response induced by CCHFV.

**KEYWORDS** CCHFV, Crimean-Congo hemorrhagic fever virus, diagnosis, glycoproteins, multiplex assay, nucleocapsid protein

Address correspondence to Patricia Sastre, patricia.sastre@eu.goldstanddiagnostics.com.

The authors declare no conflict of interest.

Crimean-Congo hemorrhagic fever (CCHF) is a tick-borne zoonotic disease caused by Crimean-Congo hemorrhagic fever virus (CCHFV), a negative-sense single-stranded RNA virus belonging to the *Bunyavirales* order. CCHF is one of the most important tick-borne viral diseases of humans and one of the most widespread arboviral diseases (1), covering a geographic area from western China to the Middle East and southeastern Europe and throughout most of Africa (2), and was more recently found in Spain (3). This geographic range is due to the wide distribution of *Hyalomma* ticks (hard ticks), the main vector of CCHFV (4). *Hyalomma* ticks infest a wide spectrum of different wildlife species (e.g., deer and hares) and free-ranging livestock animals (e.g., goats, cattle, and sheep). Hard ticks are the natural reservoir of CCHFV, and they remain infected throughout their lifetime. In addition to tick exposure, CCHF can also result from contact with the body fluid of infected animals, and human-to-human transmission can occur, as nosocomial outbreaks have been reported (5). The disease is asymptomatic in infected animals but can develop into severe illness in humans, with case-fatality rates ranging between 5 and 40% but which can be as high as 80% in some outbreaks (6). Finally, there are some concerns that climate change and warmer climates in central and northern Europe might permit the expansion of the geographic range of infected ticks by migratory birds or international animal trade (including legal, illegal, and wild animal trade) and thus expand CCHFV's range (2). Neither an effective treatment nor a vaccine has been developed against CCHFV, with ribavirin usually being used to treat patients infected with CCHFV, but with mixed results (7), or favipiravir, which has been use *in vitro* and *in vivo* with promising results (8).

The genome of CCHFV is divided into three segments, the small (S), medium (M), and large (L) segments, each encoding structural proteins. The S segment encodes the nucleocapsid (N) protein and possibly a nonstructural protein (NSs) encoded by the positive sense of the S segment (9, 10). A major role of the N protein is to encapsidate the viral RNA to form the ribonucleoprotein (RNP) complex (11). The N protein is one of the main immunogenic proteins of CCHFV and has been used in enzyme-linked immunosorbent assays (ELISAs) to detect virus-specific immunoglobulin M (IgM) and G (IgG) (12–14). The M segment encodes the glycoprotein precursor (GPC), and the L segment encodes the RNA-dependent RNA polymerase (15). The GPC is processed into two structural proteins, $G_N$ and $G_C$, and yields nonstructural secreted proteins such as GP38, GP85, and GP160 and a nonstructural protein, $NS_M$ (16, 17). $G_N$ and $G_C$ form a heterodimer which forms spikes on the envelop of the virus and mediate virus attachment and entry (15). During infection, antibodies to both proteins are produced; however, neutralizing antibodies are induced by $G_C$ and not by $G_N$ (18). The cleavage of $PreG_N$ into $G_N$ and the nonstructural proteins (GP38, GP85, and GP160) was shown to impact the replication of CCHFV (19). The role of these nonstructural proteins has not yet been elucidated, but GP38 was shown to be highly secreted by infected cells, and a monoclonal antibody (mAb) targeting this protein could protect mice from a lethal challenge (20), and its crystal structure has recently been solved (21).

Until effective treatments or vaccines are developed against CCHFV, surveillance programs are critical in order to monitor and control the spread of the virus, preventing spill-over to humans. In addition, the WHO has pointed out that diagnostic gaps for CCHFV still exist, as there are no assays that cover a range of CCHFV antigenic targets and that could potentially be used as a DIVA (differentiating infected from vaccinated animals) assay for vaccine evaluation (22). In the present work, immunogenic proteins of CCHFV, N protein, $G_N$ ectodomain ($G_N$e), and GP38 were recombinantly produced and used to develop a multiplex assay using the Luminex platform, able to simultaneously detect antibodies against all three proteins. After its development, the triplex assay was used to assess the antibody response in naturally infected animals. The triplex assay successfully detected antibodies against the three CCHFV proteins; however, differences were observed in the antibody response. All the sera had detectable antibody levels against the N protein, but only less than half of the sera had detectable antibodies against the glycoproteins. This is the first report of the simultaneous detection of antibodies against three CCHFV proteins.

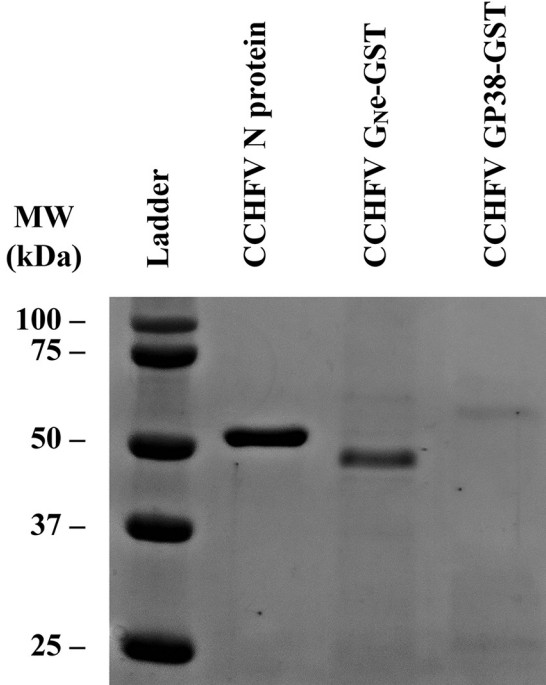

**FIG 1** SDS-PAGE analysis followed by Coomassie staining of the three purified proteins of CCHFV. From left to right: ladder (lane 1), CCHFV N protein (lane 2; molecular weight [MW], 54 kDa), CCHFV $G_N$e-GST (lane 3; MW, 46.2 kDa), and CCHFV GP38-GST (lane 4; MW, 57.8 kDa).

This multiplex assay can be a valuable tool for detecting antibodies to different CCHFV proteins, contributing to studies of the immune response to CCHFV in infected animals related to the evaluation of vaccine candidates. In addition, this assay has potential for use in surveillance programs aiming to control and prevent the further spread of the virus.

## RESULTS

**Production of recombinant proteins.** Bacterially expressed CCHFV N protein was purified by immobilized metal affinity chromatography followed by size exclusion chromatography. In contrast, the two CCHFV glycoproteins were produced in insect cells using the baculovirus expression system, with each appended to a glutathione *S*-transferase (GST)-tag, allowing their purification by affinity chromatography with an anti-GST mAb. The expression and purification of the proteins was followed by gel electrophoresis and Coomassie blue staining, which revealed pure proteins of expected molecular weights of 54 kDa, 46.2 kDa, and 57.8 kDa for CCHFV N protein, CCHFV $G_N$e, and CCHFV GP38, respectively (Fig. 1, lanes 2 to 4). The band for CCHFV GP38 is faint due to low expression and purification yields. To confirm the identity of the proteins, Western blotting (WB) was performed using 2G10, a specific anti-CCHFV N protein mAb (Fig. 2A), and 3H11, an anti-GST mAb (Fig. 2B). The results show a band corresponding to the molecular weight of CCHFV N protein (Fig. 2A, lane 2) and a band corresponding to the molecular weight of GST-tagged CCHFV $G_N$e (Fig. 2B, lane 2) and GST-tagged GP38 (Fig. 2B, lane 3). These results confirm the successful production and purification of the three CCHFV proteins. Additional bands around 25 kDa could be observed for the GST-tagged proteins (Fig. 2B), which corresponds to the molecular weight of the GST tag alone. Finally, to confirm the recognition of the recombinant proteins produced by CCHFV antibodies, WB analysis was performed using a pool of CCHFV-positive field sera. A specific band for each protein was observed (Fig. 3) with strong bands obtained for CCHFV N protein (lane 2) and CCHFV GP38 (lane 4) and a fainter band for CCHFV $G_N$e (lane 3). As expected,

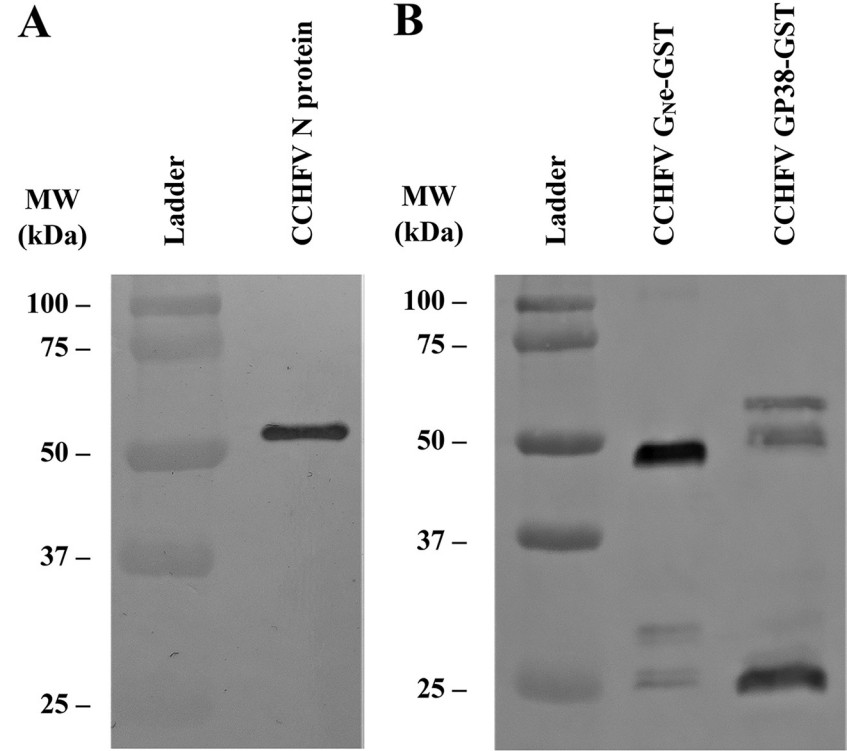

**FIG 2** Western blotting (WB) of the three purified proteins of CCHFV. (A) WB with an anti-CCHFV N protein mAb (2G10) with the recombinant CCHFV N protein (lane 2; MW, 54 kDa). (B) WB with an anti-GST mAb (3H11) with the recombinant CCHFV $G_N$e-GST (lane 2; MW, 46.2 kDa), and CCHFV GP38-GST (lane 3; MW, 57.8 kDa). For both WB, lane 1 corresponds to the ladder.

the bands corresponding to the molecular weight of the GST tags observed in Fig. 2 could not be observed in this WB.

**Optimal serum dilution.** First, the anti-CCHFV N protein and anti-GST mAbs were used to confirm the coupling of each individual antigen to its bead region and then to optimize the coupling concentration of each antigen (data not shown). The optimal protein coupling concentration was established as the highest median fluorescence intensity (MFI) obtained with the minimum amount of protein. The following quantities were used to coat $1 \times 10^6$ beads for each bead region: 25 $\mu$g of CCHFV GP38-GST (region no. 15), 50 $\mu$g of CCHFV $G_N$e-GST (region no. 20) and 25 $\mu$g of CCHFV N protein (region no. 25).

Positive and negative serum samples were used to establish the optimal assay conditions for the screening. A mix of the 3 bead regions coupled to the CCHFV proteins was incubated with serial dilutions of these positive and negative sera, and the assay was performed as described in Materials and Methods. For the positive serum samples, the same pattern of results was obtained for each bead region with a titration of the serum samples from the first dilution at 1:100 until 1:3,200, and the highest MFI was observed for the 1:100 dilution (Fig. 4A to C). For screening purposes, a dilution of the serum at 1:100 was selected (corresponding to a sample volume of 1 $\mu$L), since this was the dilution showing the highest positive/negative ratio for the three beads coated with the recombinant proteins and the highest signal with the three bead regions with the positive serum samples.

**Triplex assay.** After establishing the screening conditions, a total of 176 field serum samples comprising 29 positive ruminant field serum samples and 147 negative field serum samples, classified as positive or negative by commercial ELISA, were tested in the triplex assay to check for the presence of antibodies against the three CCHFV recombinant proteins (Fig. 5). A cutoff value was established for each bead region. The cutoff was calculated as the mean MFI obtained for the 147 negative field samples plus two standard deviations. Thus, cutoff values of 692.1 MFI, 757.5 MFI, and 6,868.5 MFI

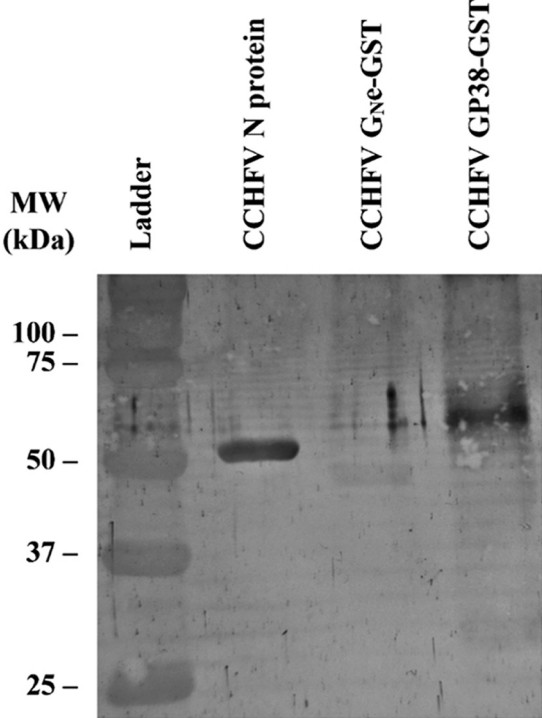

**FIG 3** Confirmation of the detection of the CCHFV recombinant proteins (N protein, $G_N$e, and GP38). Western blot analysis of the CCHFV N protein, CCHFV $G_N$e-GST, and CCHFV GP38-GST with a pool of CCHFV-positive sera as the primary antibody. A total of 0.5 $\mu$g of each protein was loaded per well. From left to right: ladder (lane 1), CCHFV N protein (lane 2; MW, 54 kDa), CCHFV $G_N$e-GST (lane 3; MW, 46.2 kDa), and CCHFV GP38-GST (lane 4; MW, 57.8 kDa).

were obtained for bead no. 15 CCHFV GP38-GST, bead no. 20 CCHFV $G_N$e-GST, and bead no. 25 CCHFV N protein, respectively. With these cutoff values, out of the 29 positive field sera, as classified by ELISA, 12 were considered positive for CCHFV GP38 or CCHFV $G_N$e (41%) and the 29 were considered positive for CCHFV N protein (100%) (Fig. 5A to C, respectively). Moreover, out of the 12 samples positive for CCHFV GP38 and $G_N$e, 11 were positive for both glycoproteins, with only 1 positive sample differing between these two proteins. Out of the samples positive for the glycoproteins, 7 of them came from cows and 5 from sheep. None of the 4 goat samples, positive for the N protein, were recognized as positive for a glycoprotein by this assay. However, using these established cutoffs, out of the 147 negative sera tested, 3 were positive for CCHFV GP38 (2%), 8 were positive for CCHFV $G_N$e (5.4%), and 5 were positive for CCHFV N protein (3.4%). The three negative field samples positive for CCHFV GP38 were also positive for CCHFV $G_N$e.

## DISCUSSION

The surveillance of diseases affecting livestock is of paramount importance for both animal and human health, especially in cases of zoonoses such as CCHF, to control the distribution of the corresponding viruses and prevent future outbreaks. Moreover, global (including climate) changes such as the increase in human and animal movements, raise the risk of spreading CCHFV into previously unaffected countries (4). Taking into account the severity of the disease and not having effective prophylaxis or treatment, implementation of surveillance strategies is critical, especially for the countries or regions with an established presence of the competent vectors (*Hyalomma* ticks) and where CCHFV has not been confirmed, yet (23, 24). Furthermore, in the WHO research and development roadmap for CCHFV diagnostics, a gap identified by the experts was to "ensure tests under development cover a range of CCHFV antigenic targets to enable their use in the assessment of live attenuated vaccines and/or as confirmatory diagnostic tests" (22).

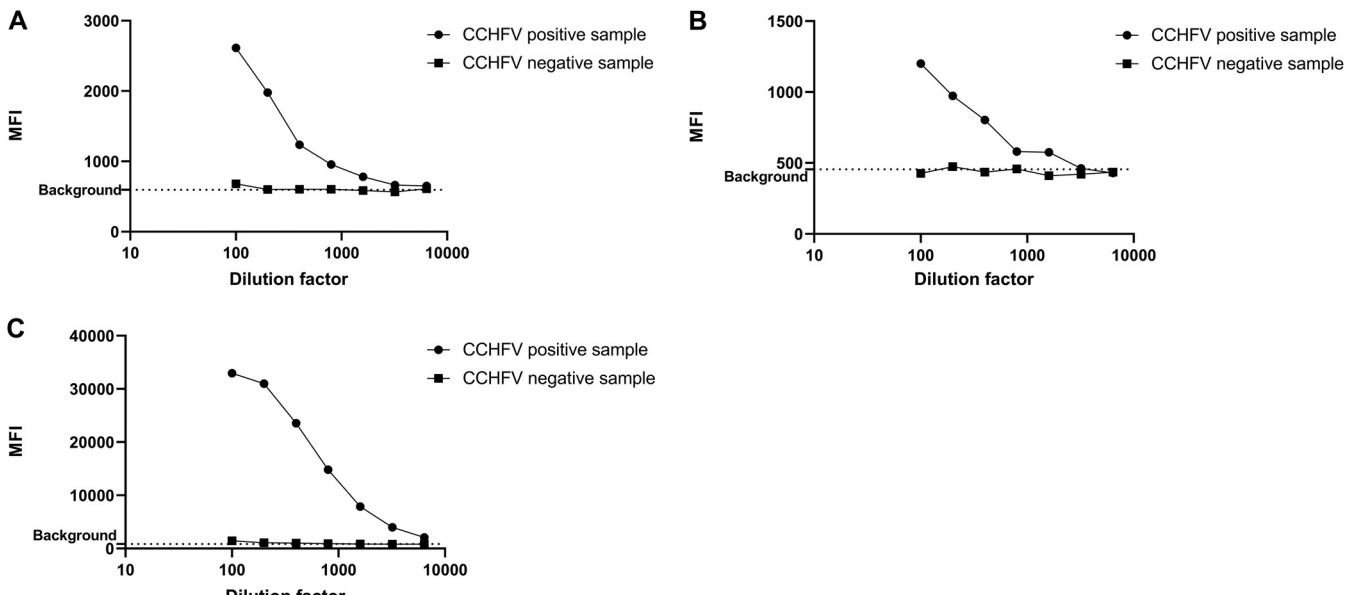

**FIG 4** Determination of the screening conditions for the triplex assay. The MFI for each bead region is given for different field serum dilutions (one positive serum sample and one negative serum sample). (A) Positive and negative field samples with bead no. 15 CCHFV GP38-GST. (B) Positive and negative field samples with bead no. 20 CCHFV $G_N$e-GST. (C) Positive and negative field samples with bead no. 25 CCHFV N protein. The signal was measured as the MFI of at least 50 events of each bead region. MFI, median fluorescence intensity.

Indeed, most of the diagnostics developed to detect antibodies against CCHFV use the CCHFV N protein (14, 25). These ELISAs show that the N protein of CCHFV is immunogenic, although it is well conserved within the *Nairoviridae* family, which could lead to some cross-reactivity (26). Some assays have been developed with other immunogenic

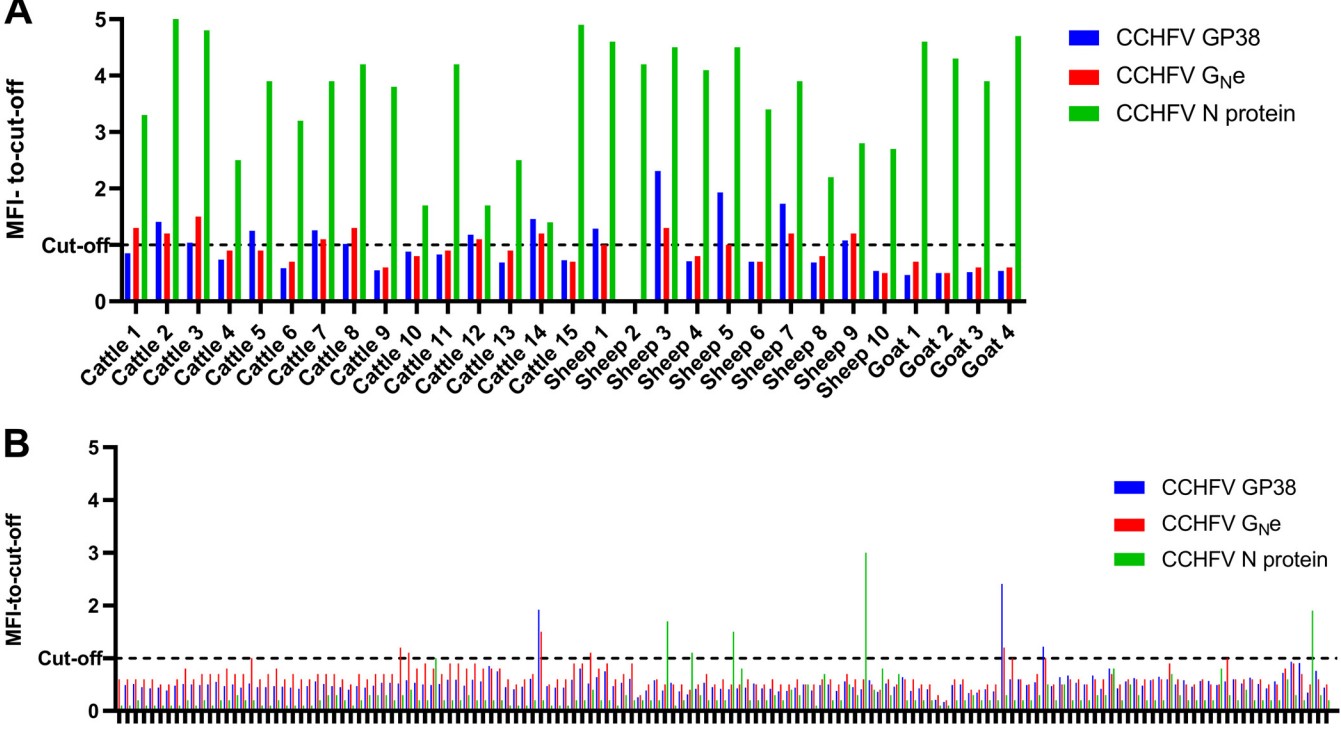

**FIG 5** Screening of the positive and negative field sera for antibodies against CCHFV GP38, CCHFV $G_N$e, and CCHFV N protein in the CCHFV triplex assay. (A) Median fluorescence intensity-to-cutoff ratio for the positive field samples. (B) Median fluorescence intensity-to-cutoff ratio for the negative field samples. The dashed line corresponds to the cutoff which was calculated as the mean obtained for the negative field samples plus two standard deviations. The signal was measured as the MFI of at least 50 events of each bead region. MFI, median fluorescence intensity.

proteins of bunyavirales members such as the $G_N e$ of RVFV in a duplex assay with the RVFV N protein (27).

In this study, three CCHFV proteins (N protein, $G_N e$, and GP38) were expressed recombinantly and then used to develop a multiplex assay to detect antibodies against these proteins in ruminant sera.

The ectodomain of $G_N$ was produced instead of the whole $G_N$ to increase the solubility of the protein compared to the whole protein with its transmembrane domain (18). Although the N protein and the $G_N e$ of CCHFV have been previously produced (28, 29), this is the first description of the recombinant expression of GP38 in insect cells that was expressed with GST as a fusion partner. The two bands observed for the GP38-GST (Fig. 2B) around the 50-kDa line probably correspond to a glycosylated and a nonglycosylated form of GP38-GST or a partial degradation of GP38 during the purification process. In reducing conditions, the N protein, $G_N e$, and GP38 were recognized by CCHFV-positive ruminant sera. However, under these conditions, the N protein and the GP38 were better recognized by CCHFV-positive sera than the $G_N e$, as shown by their higher intensity on the WB analysis (Fig. 3). This lower recognition of $G_N e$ by pooled sera is possibly due to a lower immunogenicity of $G_N e$ during natural infection, as demonstrated by the low MFI signals obtained for $G_N e$ compared to the other two proteins in the triplex assay (Fig. 5). The low recognition observed in the WB analysis could also be explained if the naturally produced antibodies were conformational rather than linear and dependent on the proper glycosylation of $G_N e$. As glycosylation obtained in insect cells is different from that obtained in mammalian cells, it could affect the reactivity of naturally produced antibodies directed to the $G_N e$. To produce recombinant CCHFV glycoproteins with posttranslational modifications closer to the native proteins, the CCHFV $G_N e$ and GP38 could be produced in mammalian cells. Other factors such as animal age, phase of the infection, time between exposure or different localities, and different decay rates of the antibody responses could also explain the lower antibody response to $G_N e$ compared to GP38 and the N protein.

To address some of the needs identified above by the WHO, a multiplex assay has been developed with different CCHFV antigenic targets (N protein, $G_N e$, and GP38 of CCHFV) in order to detect antibodies against these three proteins in ruminant sera, and the results were compared with those of a commercial ELISA, used as a reference test in the current study. One of the advantages of this assay compared to the reference assay is the volume of serum necessary to perform the triplex assay (1 $\mu$L) compared to the reference ELISA (30 $\mu$L). The immune response and, more specifically, the antibody response against CCHFV in farm animals needs to be further investigated. It is not yet known whether different farm animal species develop specific anti-CCHFV N protein, anti-CCHFV GP38, and anti-CCHFV $G_N e$ antibodies, the time points postinfection at which these can be detected and the persistence of these antibodies. The triplex assay allowed the simultaneous detection of antibodies against these three proteins in different ruminant species. Out of the 29 sera positive sera for CCHFV, confirmed by a commercial ELISA detecting antibodies against the N protein, all had antibodies against the N protein detected by the triplex assay. However, only 40% of the animals positive for the N protein had antibodies against the $G_N e$ and the GP38. The low rate of animals positive for the N protein and a glycoprotein could be explained by a higher immunogenicity of the N protein or a longer persistence of antibodies to the N protein compared to the two glycoproteins tested. As previously stated, there are still diagnostic gaps for CCHFV, and there is an absence of commercially available ELISAs based on CCHFV glycoproteins. Thus, the results obtained here with the glycoproteins could not be confirmed by commercial ELISAs. Some previous studies showed that the $G_N e$ and the GP38 elicited an immune response (20); however, here, not all the animals having antibodies against the N protein had detectable antibodies against the $G_N e$ and the GP38. One cow and two sheep originating from a Spanish farm free of diseases were positive in the triplex assay for both GP38 and $G_N e$ and negative in both the reference assay and the triplex assay for the N protein. These samples could be false-positive results; however, CCHFV has been

reported as circulating in Spain, and endemic human cases have already been reported (3). As mentioned previously, the immune response of animals to CCHFV is not known, nor is the persistence of antibodies to CCHFV antigens. However, a study from Emmerich et al. suggests that during the acute phase of CCHFV infection, antibodies are first raised against CCHFV antigens such as the envelope glycoprotein, before being raised against CCHFV N protein (30). These hypotheses should be assessed for animals as well. Overall, when comparing the two glycoproteins produced, both in reducing and nonreducing conditions, the recombinant GP38 seemed to be better detected than the recombinant $G_N$e. Moreover, this assay confirms that the N protein of CCHFV is a good candidate for serological assays, as it exhibited a higher MFI than the two glycoproteins. This hypothesis should be confirmed with human sera positive for CCHFV to see if the same trend can be observed in humans. By replacing the secondary anti-ruminant antibody used, this assay could be adapted to human sera to test the reactivity of human sera positive for these proteins and confirm the detection of CCHFV antibodies by these proteins in human sera.

As multiplex assays allow the detection of different targets simultaneously in a single reaction, other CCHFV proteins could be included in the assay, such as CCHFV $G_C$, $NS_S$, and $NS_M$. Such an assay could be used to detect antibodies against all the CCHFV proteins in the same assay and to develop a DIVA assay depending on the CCHFV vaccine candidates. In addition, other targets from different pathogens could be included in the assay for the simultaneous surveillance of multiple pathogens (31). Multiplex assays reduce the time, labor, and sample volume requirements compared to individual ELISAs and could be very useful for surveillance studies, to control the spread of viruses and prevent future outbreaks, and to better understand the immune response induced by certain viruses.

## MATERIALS AND METHODS

**Production of the recombinant nucleocapsid protein of CCHFV.** The production of CCHFV N protein was described recently (32). Briefly, cDNA designed to express the full-length N protein of CCHFV strain Baghdad 12 (GenBank accession CAD61342.1, but with conservative substitutions T111I, R195H, and H445D) was generated synthetically (Genewiz). The sequence was cloned into the pET-28a-6×His-SUMO plasmid (Thermo Fisher Scientific) for bacterial expression of the corresponding fusion protein with the small ubiquitin-like modifier (SUMO) and 6X histidine-tag at its amino terminus. The plasmid was verified by sequence analysis and transformed into *Escherichia coli* BL21(DE3) Rosetta2 (Novagen). The recombinant N protein was purified using Ni-NTA affinity chromatography followed by size exclusion chromatography, as previously described (28). In brief, the expression of CCHFV N protein was induced with 500 $\mu$M isopropyl-$\beta$-ᴅ-thiogalactoside overnight (o/n) at 18°C. The cells were harvested by centrifugation and lysed with lysozyme (1 mg/mL) and sonication. The soluble fraction was separated from the cell debris by centrifugation, after which the 6×His-SUMO-CCHFV-N was purified from the soluble fraction using Ni-NTA resin (ABT) and eluted with increasing concentrations of imidazole. The purified fusion proteins were cleaved o/n using SUMO protease (produced in-house at the University of Leeds), and the 6X histidine and SUMO tags were removed using a second nickel column, followed by a final size exclusion chromatography step with a HiLoad 26/600 Superdex 75-pg column (GE Healthcare) using an AKTA Prime system (GE Healthcare). The purification of the CCHFV N protein and its purity were assessed by SDS-PAGE followed by Coomassie staining. This purification was also followed by Western blotting with an anti-CCHFV N protein mAb (2G10) developed at Eurofins-INGENASA S.A. (32). Finally, to confirm the detection of the recombinant CCHFV N protein, a WB with CCHFV-positive field sera was performed.

**Production of the recombinant glycoproteins ($G_N$ ectodomain and GP38) of CCHFV.** cDNAs representing the CCHFV $G_N$e ($G_N$e nucleotides 1650 to 2618) and the CCHFV GP38 (nucleotides 834 to 1649) optimized for expression in *Spodoptera frugiperda* were synthetically produced (IDT), based on Nigeria/IbAr10200/1970, accessible from GenBank (AF467768.2) (16). The sequences were cloned into the cloning vector pCR8/GW/TOPO (Life Technologies) following the manufacturer's instructions. The genes were then subcloned into the transfer vector pAcSecG2T (containing a GST DNA sequence to act as a fusion partner to the expressed protein) for the protein expression in insect cells. Recombinant baculoviruses were generated as described by Hurtado et al. (33). Briefly, Sf9 insect cells cultured in Grace's medium (Life Technologies) supplemented with 8% fetal bovine serum, 0.2% pluronic acid (Kolliphor P 188, Sigma-Aldrich), and 50 $\mu$g/mL of gentamicin were cotransfected with linearized BacPak6 DNA (Clontech) and the recombinant pAcSecG2T-CCHFV $G_N$e and pAcSecG2T-CCHFV GP38, respectively, using JetPEI (Polyplus-transfection) as a transfection agent. A few days later, plaque assays were performed with the supernatant of the transfections to differentiate and select the recombinant baculoviruses by blue-white selection.

For the production of GST-tagged CCHFV $G_N$e and CCHFV GP38, Sf21 cells were infected with the corresponding recombinant baculoviruses for 96 h at 27°C. The cells were harvested by centrifugation at 1,590 $\times$ g for 10 min at 4°C, and the supernatant was kept. The supernatants were purified by affinity chromatography with NHS-activated Sepharose 4 fast-flow (GE Healthcare) beads coupled to an anti-GST mAb (3H11) produced at Eurofins-INGENASA S.A. The supernatant was added to the resin, and the

mix was incubated o/n at 4°C with gentle agitation. The nonbinding material was collected, and the resin was washed with phosphate-buffered saline (PBS). The CCHFV $G_N$e and CCHFV GP38 were eluted by addition of 0.1 M glycine-HCl, pH 2.6. The elution fractions were neutralized with 3 M Tris-HCl, pH 10, until the pH reached 7 and dialyzed o/n in PBS. The purification of the CCHFV $G_N$e and CCHFV GP38 and their purity were assessed by SDS-PAGE followed by Coomassie staining. Purification was also followed by WB with the mAb 3H11. Finally, to confirm the recognition of the recombinant glycoproteins by sera containing antibodies against CCHFV, WB with CCHFV-positive field sera was done.

**Serum samples.** For detection of antibodies to CCHFV, a collection of 29 positive serum samples comprising 15 field serum samples from cattle, 10 from sheep, and 4 from goats naturally infected by CCHFV, originating from different Macedonian farms, were provided by the Faculty of Veterinary Medicine in Skopje (FVMS, North Macedonia). A collection of 147 negative field samples (63 from cattle, 44 from goats, and 40 from sheep) from Spanish farms were evaluated. Since there are no commercial ELISAs based on CCHFV glycoproteins, the samples were classified as positive or negative based on a commercial ELISA to the N protein: ID Screen CCHF double-antigen multispecies (IDvet). This assay was used as the reference technique in the present study.

**Coupling of target antigens to beads.** The three target proteins were covalently coupled to three different bead sets (corresponding to regions no. 15, no. 20, and no. 25) of internally labeled carboxylated magnetic microspheres (Luminex) using a modified protocol of the *xMAP Cookbook* by Angeloni et al. (34). CCHFV GP38 was coupled to region no. 15, CCHFV $G_N$e to region no. 20, and CCHFV N protein to region no. 25. Briefly, $1 \times 10^6$ microspheres, identified individually by a unique fluorescence ratio, were activated by addition of sulfo-*N*-hydroxysuccinimide and 1-ethyl-3-(3-dimethylaminopropyl)carbodiimide hydrochloride, which is based on a two-step carbodiimide reaction (35). Once activated, the beads were incubated with different amounts of the corresponding protein ranging from 12.5 $\mu$g to 75 $\mu$g per one million beads in a final incubation volume of 500 $\mu$L and incubated for 2 h with rotation in the dark. After washing, the beads were blocked with PBS and 10 mM imidazole, and finally, the beads were resuspended in 1 mL of storage buffer (PBS with 1% bovine serum albumin (BSA) and 0.05% azide) and were kept in the dark at 4°C. The concentration of the beads was determined by counting on a Neubauer chamber.

**Confirmation assay and optimization.** Serial dilutions of mAbs specific to each protein, or to its GST tag, were used to perform a confirmation assay in order to assess the coupling efficiency. The mAbs were produced at Eurofins-INGENASA S.A.: 2G10 anti-CCHFV N protein for the CCHFV N protein and 3H11 anti-GST for CCHFV GP38 and $G_N$e (32). After the confirmation assay, these mAbs were used to optimize the coupling concentration of each antigen to its corresponding bead region. Then, positive and negative field sera were used to establish the screening conditions and check the reactivity of the coated beads.

**Bead-based assay for CCHFV antibody detection in ruminant sera.** The assay conditions have been described recently (32) with some modifications to reduce the background of the assay. Plates (96 wells; Stripwell microplate medium binding polystyrene, Costar) previously blocked for 30 min with StabilZyme SELECT stabilizer (Surmodics) were used for the assay. The three antigen-coupled microspheres were resuspended by vortexing and sonication in order to perform the triplex assay. A mix of beads was prepared by mixing the three bead regions in assay buffer (PBS with 5% milk and 0.3% Tween 20) to a final concentration of 25 beads/$\mu$L for each bead region. As the beads are light-sensitive, they were protected from light during the whole process. To 50 $\mu$L of individual ruminants' serum samples, diluted at 1:50 in assay buffer, was added 50 $\mu$L of the bead mixture. This mix was incubated for 30 min at room temperature (RT) and 650 rpm in a minishaker PSU-2T (Biosan). After every incubation step, a magnetic plate separator (Luminex) was used to wash the plate three times with washing buffer (PBS with 0.3% Tween 20). Then, 50 $\mu$L of the anti-ruminant monoclonal antibody EG5 labeled with biotin (Eurofins-INGENASA S.A.) was added to each well, at a final concentration of 0.5 $\mu$g/mL in assay buffer, for 30 min at RT and 650 rpm. Next, 50 $\mu$L of streptavidin R-phycoerythrin (Life Technologies) at 2 $\mu$g/mL in assay buffer was added per well and incubated for 30 min at RT and 650 rpm. Finally, the beads were washed three times with washing buffer and resuspended in 100 $\mu$L/well of washing buffer. The results were read using a MAGPIX reader (Luminex). The signal was measured as the median fluorescence intensity (MFI) of at least 50 events of each bead region. One well per assay was incubated in the absence of sample, with only dilution buffer, as blank signal. Stable and consistent blank signals were obtained throughout the development of the assay. Positive and negative controls were included in all assays to confirm the performance of the test.

**Data processing.** To statistically analyze the data obtained, MedCalc 10 software was used. The mean of the negative field samples and their standard deviation were calculated. A cutoff was established corresponding to the mean of the negative samples plus two standard deviations. Samples above this cutoff were considered positive in this assay.

**Ethical approval.** The authors confirm that the ethical policies of the journal, as noted on the journal's author guidelines page, have been adhered to and that the appropriate ethical review committee approval has been received.

## ACKNOWLEDGMENTS

We thank Isabel García and Mercedes Montón for technical assistance. We thank the EU Marie Skłodowska-Curie Actions (MSCA) Innovative Training Network (ITN): H2020-MSCA-ITN-2016, under grant no. 721367 (to Alexis C. R. Hoste) and the MediLabSecure Project, supported by the European Commission (DEVCO: IFS/2018/402-247) (to Igor Djadjovski and Miguel Ángel Jiménez-Clavero).

We declare that we have no competing interests.

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
