## [Reviewer comments · Microbiology Spectrum]

Microbiology Spectrum

Multiplex assay for simultaneous detection of antibodies against Crimean-Congo hemorrhagic fever virus nucleocapsid protein and glycoproteins in ruminants

Alexis Hoste, Igor Djadjovski, Miguel Angel Jiménez-Clavero, Paloma Rueda, John Barr, and Patricia Sastre

Corresponding Author(s): Patricia Sastre, Ingenasa

Review Timeline:

Submission Date:	July 7, 2022
Editorial Decision:	September 25, 2022
Revision Received:	December 1, 2022
Accepted:	February 4, 2023

Editor: Daniel Perez

Reviewer(s): The reviewers have opted to remain anonymous.

Transaction Report:

DOI: <https://doi.org/10.1128/spectrum.02600-22>

September 25, 2022

Dr. Patricia Sastre
Ingenasa
Av. De la Institución Libre de Enseñanza, 39
Madrid
Spain

Re: Spectrum02600-22 (Multiplex assay for simultaneous detection of antibodies against Crimean-Congo hemorrhagic fever virus nucleocapsid protein and glycoproteins in ruminants)

Dear Dr. Patricia Sastre:

Thank you for submitting your manuscript to Microbiology Spectrum. Please note that there are comments for a single reviewer as securing a second reviewer on a more timely manner was impossible and I did not want to delay the review back to you. When submitting the revised version of your paper, please provide (1) point-by-point responses to the issues raised by the reviewers as file type "Response to Reviewers," not in your cover letter, and (2) a PDF file that indicates the changes from the original submission (by highlighting or underlining the changes) as file type "Marked Up Manuscript - For Review Only". Please use this link to submit your revised manuscript - we strongly recommend that you submit your paper within the next 60 days or reach out to me. Detailed instructions on submitting your revised paper are below.

Link Not Available

Sincerely,

Daniel Perez

Journals Department
Reviewer comments:

Reviewer #1 (Comments for the Author):

This manuscript Spectrum-02600-22 by Hoste et al. describes the development of a multiplex serological assay allowing to simultaneously detect antibodies against different antigens of CCHFV in the ruminant serum. The first part details (1) the antigen production, in E.coli for the Nucleoprotein, in SF9 insect cells for two Glycoprotein segments matching Gn and Gc; (2) the coating of the recombinant antigens on beads for multiplex analysis. In the second part, the triplex assay is evaluated on a total of 176 ruminant field serum samples including 29 «positive» according to the IDVet CCHF Double Antigen Multi-species kit and 147 negative field serum samples.

It is a matter of fact that CCHFV serology is currently under rapid evolution with several kits developed (in particular by IDVet) trying both to define the right line for positivity (ruminants over the world have not the same serological background) and an easy use for various animal species. The principal interest of the present manuscript is clearly to allow a parallel analysis of the serological response to 3 main viral antigens (both N and Gs). However, the present form is clearly incomplete and perfectible. I hope that the following remarks will allow to improve the interest for this work.

Antigens production and coating

- General remark : CCHFV displays a large diversity and this may influence the serological response. Is it possible to clarify why the Bagdad strain has been used for the N and the Ibadan strain for the Gs.
- Lines 173-4 : the Gn and GP38 are purified through a step at a very acidic pH 2.6 before neutralisation at pH 10. Assuming that the virus-cell envelope-membrane fusion preceding the RNP entry into the cytoplasm necessitates a conformational change of G under acidic pH, is it possible that such pH treatment would interfere with the G antigens conformation and therefore on their recognition by serum antibodies.
- Figure 2B : is it possible to discuss the two bands observed for the GP38-GST. Only the GST alone is discussed.
- Figure 3 : while fig 1 and 2 repeatedly suggest the GP38 is expressed in lower amounts than Gn, the WB with the pool of positive sera clearly shows the inverse. Despite the fact that the following steps are probably equilibrating the response (50µg of Gn, 25µg for N and G38), this is an important information which has to be linked to the previous remark about possible conformational modifications provoked by the purification treatment. Is Gn as correctly folded than GP38 and N ? Is this interfering with the further serological results ? This observation is outlined by authors on lines 404-06. Their explanations proposed in lines 375-84 (less Gn immunogenicity or different glycosylation) are possible but may be not the only ones.
- Lines 192-3 and later in the text : "The three target proteins were covalently coupled to three different regions 192 (regions #15, #20, #25) of carboxylated magnetic microspheres (Luminex)". This sentence is not clear to me (as a reader) even if a reference is given. Different beads of different colours for each antigens? Different region of each bead!!!! Clarify please.
- Figure 4 : the panels and writings are very small.

Triplex assay, Figure 5

- The ID Screen CCHF Double Antigen Multi-species (IDVet) is used as the reference technique to decide on "CCHFV positivity". This may be discussed since it has a global tendency to give a high back-ground.
- Technically, I would suggest to put the positive samples in a different colour in order to distinguish them more easily. Also, if it is possible to join the 3 panels with very slight vertical lines (or another way) to join the identical samples, this would facilitate the understanding of the explanations given in Results and also in Discussion.
- Lines 330-2 : is it possible to discuss a possible effect of the species. Is the background the same for cow, sheep and goat ? BTW, it would be also interesting to design then on figure 5 to illustrate this possible species effect.
- It may be interesting in the discussion to develop a bit more about the IDVet "negative" ELISA samples that are positive for the 2 Gs but not with N. In which animal species ? From which country (link with current transmission to human ?). Why a response against G and not against N ?

Staff Comments:

Preparing Revision Guidelines

Please return the manuscript within 60 days; if you cannot complete the modification within this time period, please contact me. If you do not wish to modify the manuscript and prefer to submit it to another journal, please notify me of your decision immediately so that the manuscript may be formally withdrawn from consideration by Microbiology Spectrum.

Microbiology Spectrum - Spectrum-02600-22 “Multiplex assay for simultaneous detection of antibodies against Crimean-Congo hemorrhagic fever virus nucleocapsid protein and glycoproteins in ruminants”

24th November 2022

Dear Editor,

Thank you very much for revising the Manuscript Spectrum-02600-22 entitled " Multiplex assay for simultaneous detection of antibodies against Crimean-Congo hemorrhagic fever virus nucleocapsid protein and glycoproteins in ruminants". We hope to answer accordingly to all your queries.

All the corrections were done using the tracking change of MS Word and the line numbers correspond to the corrected version.

Reviewer comments:

Reviewer #1 (Comments for the Author):

This manuscript Spectrum-02600-22 by Hoste et al. describes the development of a multiplex serological assay allowing to simultaneously detect antibodies against different antigens of CCHFV in the ruminant serum. The first part details (1) the antigen production, in E.coli for the Nucleoprotein, in SF9 insect cells for two Glycoprotein segments matching Gn and Gc; (2) the coating of the recombinant antigens on beads for multiplex analysis. In the second part, the triplex assay is evaluated on a total of 176 ruminant field serum samples including 29 «positive» according to the IDVet CCHF Double Antigen Multi-species kit and 147 negative field serum samples.

It is a matter of fact that CCHFV serology is currently under rapid evolution with several kits developed (in particular by IDVet) trying both to define the right line for positivity (ruminants over the world have not the same serological background) and an easy use for various animal species. The principal interest of the present manuscript is clearly to allow a parallel analysis of the serological response to 3 main viral antigens (both N and Gs). However, the present form is clearly incomplete and perfectible. I hope that the following remarks will allow to improve the interest for this work.

Antigens production and coating

- General remark : CCHFV displays a large diversity and this may influence the serological response. Is it possible to clarify why the Bagdad strain has been used for the N and the Ibadan strain for the Gs.

We agree that CCHFV strains are genetically highly diverse, and this surely can influence the serological response. However, the N protein sequences are not very variable and the N protein sequences from Baghdad strain and Nigeria/IbAr10200/1970 are 95% identical (blast alignment on the sequences below) and the Baghdad strain used in this paper was already available in the lab. For the Gs, we have used IbAr as it was the model for CCHFV and the most studied strain for which we could determine the ectodomain of the G_{NE} and the GP38. The homology between the G_{NE} and GP38 of IbAr and Baghdad strain is 92% identical. The main difference in the identity of the M segment is mostly due to the murine-like domain.

```

Query: CAD61342.1 nucleocapsid [Crimean-Congo hemorrhagic fever orthonavirius] Query ID: lc1|Query_40243 Length: 482

>sp|P89522|NCAP_CCHFI Nucleoprotein OS=Crimean-Congo hemorrhagic fever virus (strain Nigeria/IbAr10200/1970) OX=652961 GN=N PE=1 SV=1
Sequence ID: Query_40245 Length: 482
Range 1: 1 to 482

Score:972 bits(2514), Expect:0.0,
Method:Compositional matrix adjust.,
Identities:460/482(95%), Positives:472/482(97%), Gaps:0/482(0%)

Query 1  MENKIEVNSKDEMNKWFEEFKKGNGLVDITYNSYSFCESVPLDRFVFQMGATDDAQKD 60
MENKIEVN+KDEMN+WFEFEFKKGNGLVDY+TNSYSFCESVPLDRFVFQMA ATDDAQKD
Sbjct 1  MENKIEVNNKDEMNRFEEFKKGNGLVDYFTNSYSFCESVPLDRFVFQMASATDDAQKD 60

Query 61  SIYASALVEATKFCAPIYECAMWSTGIVKKGLEWFEKNTGTIKSWDESIEYELKVEVPKI 120
SIYASALVEATKFCAPIYECAM SSTGIVKKGLEWFEKN GTIKSWDESIEY ELKV+VPKI
Sbjct 61  SIYASALVEATKFCAPIYECAMWSTGIVKKGLEWFEKNAGTIKSWDESIEYELKVDVPKI 120

Query 121  EQLFNYQQAALKWRKDIGFRVNANTAAALSNKVLAEYKVPGEIVMSVKEMLSDMIRRRNLI 180
EQL YQQAALKWRKDIGFRVNANTAAALSNKVLAEYKVPGEIVMSVKEMLSDMIRRRNLI
Sbjct 121  EQLTGYQQAALKWRKDIGFRVNANTAAALSNKVLAEYKVPGEIVMSVKEMLSDMIRRRNLI 180

Query 181  LNRGGDENPRGPVSHVWCREFVKGYIMAFNPPWGDINKSGRSGIALVATGLAKLAE 240
LNRGGDENPRGPVSHVWCREFVKGYIMAFNPPWGDINKSGRSGIALVATGLAKLAE
Sbjct 181  LNRGGDENPRGPVSHVWCREFVKGYIMAFNPPWGDINKSGRSGIALVATGLAKLAE 240

Query 241  TEGKGVFDEAKKTVEALNGYLDKHKDEVDKASADNMVNTLLKHVAKAQELYKNSSALRAQ 300
TEGKG+FDEAKKTVEALNGYLDKHKDEVD+ASAD+M+TNLLKH+AKAQELYKNSSALRAQ
Sbjct 241  TEGKGVFDEAKKTVEALNGYLDKHKDEVDKASADSMITNLLKHIAKQELYKNSSALRAQ 300

Query 301  GAQIDTVFSSYYWLYKAGVTPETFPVTSQFLFELGKHPRGTKMKMKALLSTPMKNGKKLY 360
AQIDT FSSYYWLYKAGVTPETFPVTSQFLFELGK PRGTKMKMKALLSTPMKNGKKLY
Sbjct 301  SAQIDTAFSSYYWLYKAGVTPETFPVTSQFLFELGKQPRGTKMKMKALLSTPMKNGKKLY 360

Query 361  ELFADDSFQQNRIYMHPAVLTAGRISEMGVCFGTIPVANPDDAALGSGHTKSILNLRNT 420
ELFADDSFQQNRIYMHPAVLTAGRISEMGVCFGTIPVANPDDAA GSGHTKSILNLRNT
Sbjct 361  ELFADDSFQQNRIYMHPAVLTAGRISEMGVCFGTIPVANPDDAAQSGHTKSILNLRNT 420

Query 421  ETNNPCARTIVKLFQIKTGFNIQDMDIVASEHLLHQSLVGKQSPFNAYNVKGNATSAN 480
ETNNPCA+TIVKLFQ+QKTGFNIQDMDIVASEHLLHQSLVGKQSPFNAYNVKGNATSAN
Sbjct 421  ETNNPCAARTIVKLFQIKTGFNIQDMDIVASEHLLHQSLVGKQSPFNAYNVKGNATSAN 480

Query 481  II 482
II
Sbjct 481  II 482

```

- Lines 173-4 : the Gn and GP38 are purified through a step at a very acidic pH 2.6 before neutralisation at pH 10. Assuming that the virus-cell envelope-membrane fusion preceding the RNP entry into the cytoplasm necessitates a conformational change of G under acidic pH, is it possible that such pH treatment would interfere with the G antigens conformation and therefore on their recognition by serum antibodies.

We agree that such pH treatment as described by the reviewer could interfere with the conformation of the glycoproteins recombinantly produced in this article. However, the G_{NE} and GP38 are purified through a very acidic pH 2.6 before being neutralized by addition of 3 M Tris-HCl, pH 10 until the pH reached 7, but the pH was not brought under basic conditions. The correction was made to L170, p8. We apologize if the sentence in the M&M was not clear

- Figure 2B : is it possible to discuss the two bands observed for the GP38-GST. Only the GST alone is discussed.

This point has been corrected and is discussed L372-375, p16. The two bands observed for the GP38-GST on Figure 2B around the 50 kDa line probably correspond to a glycosylated and a non-glycosylated form of GP38-GST or a partial degradation of GP38 during the purification process.

- Figure 3 : while fig 1 and 2 repeatedly suggest the GP38 is expressed in lower amounts than Gn, the WB with the pool of positive sera clearly shows the inverse. Despite the fact that the following steps are probably equilibrating the response (50µg of Gn, 25µg for N and G38), this is an important information which has to be linked to the previous remark about possible conformational modifications provoked by the purification treatment. Is Gn as correctly folded than GP38 and N ? Is this interfering with the further serological results ? This observation is outlined by authors on lines 404-06. Their explanations proposed in lines 375-84 (less Gn immunogenicity or different glycosylation) are possible but may be not the only ones.

On the SDS-PAGE and on the WB (Figure 1 and 2B), we can see that GP38 is expressed in lower amounts than G_{NE} (fainter band compared to G_{NE}). For the WB with the pool of positive sera (Figure 3), the same amount of protein was loaded per well (0.5 µg of each protein), to avoid accounting for the expression yield. On this WB, strong bands for CCHFV N protein and GP38 can be seen, meaning that they are better recognized by the antibodies present in the CCHFV pool sera than G_{NE} which shows a fainter band. We agree that this result could be due to a lower immunogenicity of G_{NE}, different glycosylation or incorrect folding of G_{NE} compared to the other two proteins tested. This lower response of G_{NE} in the WB, but also when setting up the Luminex assay, explains why we had to increase the concentration of G_{NE} when labelling the Luminex beads, to increase the signal obtained with G_{NE} in the Luminex assay. This led to signals obtained with G_{NE} close to the ones obtained with GP38 in the Luminex assay. Other factors such as animal age, phase of the infection, time between exposure or different localities, and different decay rates of the antibody responses could also explain the lower antibody response to G_{NE} compared to GP38 and the N protein. This was added L387-390, p17.

- Lines 192-3 and later in the text : "The three target proteins were covalently coupled to three different regions 192 (regions #15, #20, #25) of carboxylated magnetic microspheres (Luminex)". This sentence is not clear to me (as a reader) even if a reference is given. Different beads of different colours for each antigens? Different region of each bead!!!! Clarify please.

The corrections were made L188-192, p8 to clarify this part. The microspheres include two internal dyes where a first excitation wavelength allows the identification of each microsphere from 100 unique microsphere sets. Each antigen (N protein, G_{NE} and GP38) was coupled to a specific set of beads (#15, #20, #25, also called bead regions) which can be recognized by the equipment. Then, a second excitation wavelength allows the observation of the fluorescent

reporter molecule (in our case Streptavidin R-phycoerythrin), allowing the detection of the analyte captured on the surface of the microsphere.

- Figure 4 : the panels and writings are very small.
The corrections were made.

Triplex assay, Figure 5

- The ID Screen CCHF Double Antigen Multi-species (IDVet) is used as the reference technique to decide on "CCHFV positivity". This may be discussed since it has a global tendency to give a high back-ground.

According to the manufacturer, the ID Screen CCHF Double Antigen Multi-species has a high specificity (100%) and sensitivity (98.9 %) across multiple species.

If the reviewer means that the assay has a tendency to give a high noise-to-signal ratio, we have not observed a high noise-to-signal ratio in all the negative sera we have been testing with the ID Screen CCHF Double Antigen Multi-species. In a study performed by one of our collaborators with over 300 negative samples tested in this assay, the signal obtained was: from 0,036 to 0,059 = 297 sera (90,27%); from 0,060 to 0,1 = 22 sera (6,68%); and from 0,1 to 0,4 = 10 sera (3,29%), with no false-positive.

If the reviewer means that the assay has a tendency to give a high number of false-positives, from our side, we have tried to confirm the results obtained with the ID Screen CCHF Double Antigen Multi-species with other techniques, mainly by IFA using the animal-adapted EUROIMMUN IFA kit and in some cases, the animal-adapted Vector-Best ELISA. The IFA assay confirmed about half of the positive samples identified with the ID Screen CCHF Double Antigen Multi-species. This could indeed indicate a lack of specificity of the CCHF Double Antigen Multi-species, but also may be due to a lack of sensitivity of the IFA assay. Some positive sera (ID Screen +) giving either a positive or negative result in the IFA (IFA + or IFA -) were also tested in the Vector-Best ELISA, which confirmed the ID Screen + as true positives. In our opinion, the IFA is less sensitive than either ELISA. However, as there is no "gold standard" technique to rely on as a reference, we are always working with some level of uncertainty.

- Technically, I would suggest to put the positive samples in a different colour in order to distinguish them more easily. Also, if it is possible to join the 3 panels with very slight vertical lines (or another way) to join the identical samples, this would facilitate the understanding of the explanations given in Results and also in Discussion.

We agree and have changed the figure accordingly. All the positive samples are now regrouped on the same figure (Figure 5A) and the negative samples are regrouped in second figure (Figure 5B). For better clarity, the median fluorescence intensity was converted to a median fluorescence intensity-to-cut-off ratio. The title of Figure 5 has been changed accordingly:

Figure 5. Screening of the positive and negative field sera for antibodies against CCHFV GP38, CCHFV G_{Ne} and CCHFV N protein in the CCHFV triplex assay. A: median fluorescence intensity-to-cut-off ratio for the positive field samples. B: median fluorescence intensity-to-cut-off ratio for the negative field samples. The dashed line corresponds to the cut-off which was calculated as the mean obtained for the negative field samples plus two standard

deviations. The signal was measured as MFI of at least 50 events of each bead region. MFI: median fluorescence intensity.

- Lines 330-2 : is it possible to discuss a possible effect of the species. Is the background the same for cow, sheep and goat ? BTW, it would be also interesting to design then on figure 5 to illustrate this possible species effect.

We agree and have made the changes to Figure 5A to show the different species of the positive field samples. The background of the animals is not known as the positive samples are samples that have been collected in the field in Macedonian farms. The following sentence was added L397-401, p17 to discuss the possible effect of the species: “The immune response and more specifically the antibody response against CCHFV in farm animals needs to be further investigated. It is not yet known whether different farm animal species develop specific anti-CCHFV N protein, anti-CCHFV GP38 and anti-CCHFV G_{NE} antibodies, the time points post-infection at which these can be detected and the persistence of these antibodies”.

- It may be interesting in the discussion to develop a bit more about the IDVet "negative" ELISA samples that are positive for the 2 Gs but not with N. In which animal species ? From which country (link with current transmission to human ?). Why a response against G and not against N ?

We agree with this comment and the following sentences have been added in the discussion at L414-423, p18 linked to the previous comment by the reviewer. “One cattle and two sheep originating from Spanish farm free of diseases were positive in the triplex assay for both GP38 and G_{NE} and negative in both the reference assay and in the triplex assay for the N protein. These samples could be false positive results, however, CCHFV has been reported as circulating in Spain and endemic human cases have already been reported (Negredo et al., 2019). As mentioned previously, the immune response of animals to CCHFV is not known, nor the persistence of antibodies to CCHFV antigens. However, a study from Emmerich et al. suggests that during the acute phase of CCHFV infection, antibodies are first raised against CCHFV antigens such as the envelope glycoprotein, before being raised against CCHFV N protein (Emmerich et al., 2018). These hypotheses should be assessed for animals as well.”

February 4, 2023

Dr. Patricia Sastre
Ingenasa
Av. De la Institución Libre de Enseñanza, 39
Madrid
Spain

Re: Spectrum02600-22R1 (Multiplex assay for simultaneous detection of antibodies against Crimean-Congo hemorrhagic fever virus nucleocapsid protein and glycoproteins in ruminants)

Dear Dr. Sastre:

I apologize for the delay in handling your manuscript. It has been a rough couple of months. Your manuscript has been accepted, and I am forwarding it to the ASM Journals Department for publication. You will be notified when your proofs are ready to be viewed.

Sincerely,

Daniel Perez
Editor, Microbiology Spectrum
